# Meta-learning from relevant demonstrations improves compositional generalization

**Sam Spilsbury**
Department of Computer Science
Aalto University
Espoo, Finland
sam.spilsbury@aalto.fi

**Alexander Ilin**
Department of Computer Science
Aalto University
Espoo, Finland
alexander.ilin@aalto.fi

## Abstract

We study the problem of compositional generalization of language-instructed agents in gSCAN. gSCAN is a popular benchmark which requires an agent to generalize to instructions containing novel combinations of words, which are not seen in the training data. We propose to improve the agent's generalization capabilities with an architecture inspired by the Meta-Sequence-to-Sequence learning approach [29]. The agent receives as a context a few examples of pairs of instructions and action trajectories in a given instance of the environment (a support set) and it is tasked to predict an action sequence for a query instruction for the same environment instance. The context is generated by an oracle and the instructions come from the same distribution as seen in the training data. In each training episode, we also shuffle the indices of the attributes of the observed environment states and the words of the instructions to make the agent figure out the relations between the attributes and the words from the context. Our predictive model has the standard transformer architecture. We show that the proposed architecture can significantly improve the generalization capabilities of the agent on one of the most difficult gSCAN splits: the "adverb-to-verb" Split H.

## 1   Introduction

We want autonomous agents to have the same compositional understanding of language that humans do [10; 50]. Without this understanding, the sample complexity required to train them for a wide range of compositions of instructions would be very high [48; 26]. Naturally, such compositional generalization has received interest from both the language and reinforcement learning communities. "Compositional Generalization" can be divided into several different sub-skills, for example being able to reason about object properties compositionally [7; 39], composing sub-instructions into a sequence [31; 34] or generating novel action sequences according to novel instructions made up of familiar components [30]. In this work, we examine this latter challenge in more detail by focusing on Split H of gSCAN [42], otherwise known as the "adverb-to-verb" split.

gSCAN[1] is a testing environment for language-grounded agents, consisting of a 6-by-6 grid-world where each episode has a *state* (a unique combination of objects and initial agent position) and some language instruction. The instructions follow a template of "`action a size? color? object adverb`", where `?` indicates that a token is optional. Certain combinations of instructions and object combinations are not found in the training set. A *success* happens when the agent exactly matches the target actions for an episode. A more detailed description of gSCAN is found in Appendix A.

---

[1] https://github.com/LauraRuis/GroundedScan, MIT License

36th Conference on Neural Information Processing Systems (NeurIPS 2022).

Test Split H contains only instructions following the template "`pull a size? color? object while spinning`". This requires the agent to walk towards the target object and pull it the required number of times, while at the same time performing actions `LTURN(4)` after each `WALK` and `PULL`. These instructions from Test Split H, nor their corresponding action sequences are found in the training data. The nature of the distributuon shift is shown in Appendix B. Solving the problem requires the agent to generate the unseen action trajectory based on a compositional understanding of the instructions.

We hypothesize that a promising approach is Meta Sequence-to-Sequence Learning (meta-seq2seq) [29]. We think the reason why this approach works well is the permutations applied to the supports and target instructions, which ensures that an agent does not overfit to particular sequences of symbols in the output space and instead forces meta-learning to determine what the true output actions should be for a given episode. Extending this approach to language grounding environments was flagged as a possible future work direction in [42]. In this work we propose to do exactly that.

Our contributions are: **first**, we describe an extension of meta-seq2seq with state-relevant supports, **second** we report promising success rate performance on gSCAN Split H, **third** we explain different ways to generate the supports and how this affects performance, and **fourth** we motivate how this approach aligns with intuitions about human compositional problem solving.

## 2   Related Work

**Compositional Generalization**   There is a long line of work on the challenge of compositional generalization in deep learning. Initial works show that RNNs cannot solve these problems well [50; 32]. Datasets such as SCAN [30], COGS [27], 0gendata [15] and PCFG [25] serve as benchmarks to measure progress. Since then, various approaches have been proposed to improve compositional generalization performance, including data augmentation [1; 45; 19; 40], problem-specific inductive biases [5; 43; 20; 54; 49**?** ], increased data diversity [37; 1], transfer learning [56] and neural module networks [2; 41]. These approaches can perform very well, but usually require prior assumptions about underlying data. In computer vision and multimodal domains, the Transformer architecture has been shown to solve some compositional generalization tasks [51; 24; 8; 47]. Transformer's success on token-level tasks is also promising but still limited [38; 3; 39; 40; 47]. Meta-learning [11; 53; 35] and group equivariance [16] have also shown promise on such problems.

**Grounded Environments**   Many language grounding environments exist, such as BabyAI [9], ALFRED [46], VizDoom [7] and SILG [55]. gSCAN and its derivatives [42; 52] specifically focus on task compositional generalization in an interactive world. The various splits of gSCAN are still not completely solved. Various approaches proposed include linguistic-assisted attention [28], graph networks [14], neural module networks [22; 41], data augmentation [44], entropy regularization [**?** ] and Transformers [39]. Splits D, G and H remain challenging to solve with a general approach.

**In-context and model-based learning**   We take inspiration from a long line of work on in-context meta-learning, starting with $RL^2$ [13] for RNNs and more recently the extension to transformers with TrML [33]. Also related to this work is the idea of retrieval for in-context learning [18; 4] and proposing goals and planning in an imagination of the world [36; 6; 21; 12].

## 3   Method

The meta-seq2seq architecture and training method [29] uses token-symbol permutation and meta-learning to novel-sequence compositional generalization problems. The main idea behind meta-seq2seq is learn a model that is robust to *permutations* in the token-symbol mapping by providing *supports* of how a given permutation of symbols are used to solve other problems in the training data. The supports consist of *support instructions* $I_1, ..., I_n$ and corresponding *support targets* $A_1, ..., A_n$. A *query instruction* $I^Q$ is given and the model must predict the corresponding *query targets* in the *permuted* output space $A^Q$ for a permuted $I^Q$. We make random permutations of token-symbol mappings for **both** the instructions and targets using the Permuter block. A consistent permutation is applied for both the queries and supports. See Appendix I for examples of permutations. At testing time, no permutation is applied. The permutations make the task unsolvable without the supports, forcing the model to make use of them and also not to overfit to particular symbol sequences. Figure 1 shows our implementation and how the supports and query are encoded.

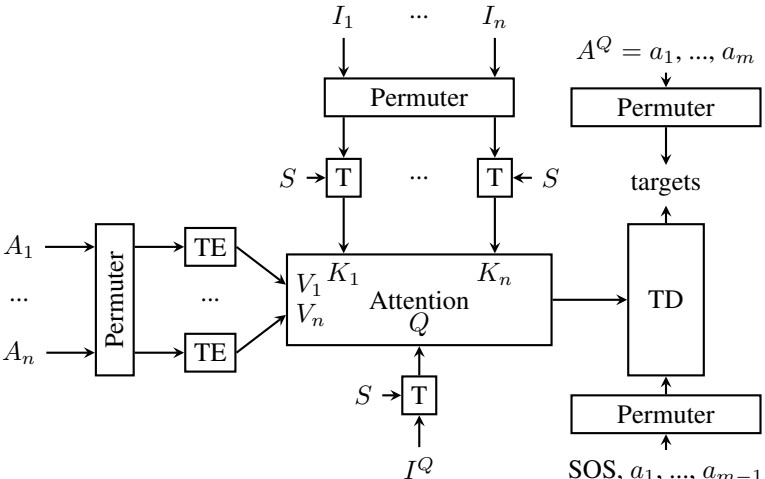

Figure 1: A schematic showing our proposed method. An oracle function, generates relevant in-distribution support instructions sequences $I_1, ..., I_n$, which contain parts of the query instruction, but not the query itself. The action generator, which could be an oracle function or an existing in-distribution model, generates the corresponding support target sequences $A_1, ..., A_n$ for those instructions in the environment. The support instructions and targets are embedded using a Transformer (T) by encoding state $S$ and decoding instruction $I_j$. Support targets are encoded with a Transformer Encoder (TE). Attention is between each word in $I^Q$ and each encoded support instruction $I_1, ..., I_n$. The output of this attention is the input sequence for a Transformer Decoder (TD), which predicts the permuted target indices for $I^Q$ using causal masking. The agent must use the supports to solve the query instruction as the word and target indices are re-permuted on every episode.

However, it has until now remained an open question how such an approach could be applied to state-conditional problems such as gSCAN [42]. The main challenge is that both meta-seq2seq and GECA [1] rely on *retrieval* from the training data. As observed in [35], this is problematic because the supports need to be relevant to the task at hand and they won't be relevant at the point where there is a mismatch between the support state and the query state.

Therefore, the key extension to meta-seq2seq in our work is to generate state-relevant supports using an oracle function. The generated instructions and actions pertain to the same object, but with different in-distribution verbs and adverbs. The oracle does not generate an example of the query instruction, nor instructions only seen in Split H. See Appendix G for a detailed description.

## 4   Experiments

We ran experiments to compare success rate performance of our approach against a transformer [39] and other baselines, shown in Table 1. Both the transformer and the adapted meta-seq2seq approach above have a similar parameter budget, effective batch size, learning rate, and training iteration count. Appendix C contains on the finer details of training and the model. Runs are over seeds 0-9 and the bottom quantile of seeds by Split A performance are excluded. In Ours(o, A) we measure validation performance on all the other splits by Split A performance, since the other splits are in-principle unseen. Ours(o) is our model at the best Split H performance

We achieve very strong performance on Split H, comparable to the recent work of Ruis and Lake [41]. An analysis of the remaining failure cases is provided in Appendix J. Note that for splits B-F, the oracle demonstrates related instructions to the corresponding to an out-of-distribution object or location, so those results are not exactly comparable to prior work, but they do demonstrate that a few-shot compositional meta-learning paradigm can extend to other splits. We also see that a Transformer at the same scale starts to have degraded performance on out-of-distribution splits, indicating that it continues to overfit to the in-distribution set as the model scales up.

Table 1: Our approach compared to other recent works on the gSCAN dataset. Numbers are mean success rate over 10 seeds ± standard deviation. Additional comparisons, can be found Appendix F.

|  | ViLBERT [39] | Modular [41] | Role-guided [28] | Transformer Ours | Ours(o, A) Ours | Ours(o) Ours |
|---|---|---|---|---|---|---|
| #params | 3M |  |  | 13.2M | 13.2M | 13.2M |
| A | 99.95 ± 0.02 | 96.34 ± 0.28 | 96.73 ± 0.58 | **1.0 ± 0.0** | 0.96 ± 0.0 | 0.95 ± 0.01 |
| B | **99.90 ± 0.06** | 59.66 ± 23.76 | 94.91 ± 1.30 | 0.91 ± 0.15 | 0.96 ± 0.01 | 0.96 ± 0.01 |
| C | **99.25 ± 0.91** | 32.09 ± 9.79 | 67.72 ± 10.83 | 0.86 ± 0.19 | 0.97 ± 0.01 | 0.97 ± 0.01 |
| D | 0.0 ± 0.0 | 0.0 ± 0.0 | 11.52 ± 8.18 | 0.0 ± 0.0 | **0.35 ± 0.06** | 0.35 ± 0.05 |
| E | **99.02 ± 1.16** | 49.34 ± 11.60 | 76.83 ± 2.32 | 0.79 ± 0.26 | 0.98 ± 0.01 | 0.97 ± 0.01 |
| F | 99.98 ± 0.01 | 94.16 ± 1.25 | 98.67 ± 0.05 | **1.0 ± 0.0** | 0.97 ± 0.01 | 0.97 ± 0.01 |
| H | 22.16 ± 0.01 | 76.84 ± 26.94 | 20.98 ± 1.38 | 0.08 ± 0.09 | 0.81 ± 0.03 | **0.86 ± 0.02** |

Table 2: Different types of oracle behaviour. Numbers are successes ± standard deviation over top 5 of 10 seeds by Split A performance, due to some seeds taking much longer to converge than others. Ablations measured at 28,000 iterations at the best validation checkpoint on Split A.

|  | Ours(o, A) | No permutations | Transformer Actions | Distractors | Irrelevant Instructions | Retrieval |
|---|---|---|---|---|---|---|
| A | 0.96 ± 0.0 | 0.97 ± 0.0 | 0.94 ± 0.01 | 0.78 ± 0.12 | 0.27 ± 0.01 | 0.23 ± 0.02 |
| B | 0.96 ± 0.01 | 0.98 ± 0.0 | 0.95 ± 0.01 | 0.82 ± 0.1 | 0.27 ± 0.02 | 0.16 ± 0.03 |
| C | 0.97 ± 0.01 | 0.98 ± 0.0 | 0.96 ± 0.01 | 0.82 ± 0.09 | 0.27 ± 0.0 | 0.01 ± 0.0 |
| D | 0.35 ± 0.06 | 0.03 ± 0.03 | 0.0 ± 0.0 | 0.17 ± 0.1 | 0.02 ± 0.02 | 0.0 ± 0.0 |
| E | 0.98 ± 0.01 | 0.98 ± 0.0 | 0.97 ± 0.0 | 0.83 ± 0.12 | 0.28 ± 0.02 | 0.04 ± 0.02 |
| F | 0.97 ± 0.01 | 0.99 ± 0.0 | 0.96 ± 0.01 | 0.79 ± 0.13 | 0.23 ± 0.02 | 0.34 ± 0.04 |
| H | 0.81 ± 0.03 | 0.17 ± 0.07 | 0.76 ± 0.03 | 0.49 ± 0.17 | 0.0 ± 0.0 | 0.08 ± 0.01 |

We also studied different kinds of oracle behaviour, shown in Table 2. Without permutations, performance on Splits D and H are comparable to a Transformer, confirming the result in [29]. **Distractors** shows that when 3 out of 8 instructions are object-irrelevant, performance is worse, but this is mainly due to higher variance in convergence rate between seeds. **Irrelevant Instructions** and **Retrieval** show that generating completely irrelevant support instructions or retrieving environment layouts for relevant instructions causes performance on all splits to drop significantly. **Transformer Actions** tests action supports being generated by a Transformer receiving oracle instructions and the state. The Transformer was trained on the gSCAN training set and had a 95% autoregressive generation success rate on Split A. Results are comparable, except on Split D.

## 5   Discussion and Conclusion

The extension of meta-seq2seq we present in this paper has promising results in the challenging Split H of the gSCAN benchmark. We believe its architecture is also simple and intuitive to understand through the lens of how humans might think about compositional problems. When faced with an unfamiliar instruction ($I^Q$), the agent thinks of similar instructions ($I_1, ..., I_n$) and their solutions ($A_1, ..., A_n$) in the same environment. The agent then thinks about how those solutions can be composed in light of the current instruction.

At present, an important limitation of this work is that an oracle function generates $I_1, ..., I_n$ and $A_1, ..., A_n$ for a given $I^Q$. We hope to extend this work by replacing the oracle with a generative model, which generates in-distribution query instructions and their corresponding actions with reference to the generative distribution of the training data. There is also a slight degradation of performance on the other splits, including the in-distribution Split A, compared to a Transformer. In light of the limitations, we suggest that applications use this work in conjunction with a baseline, for example by distillation [23] or by using this approach as a sort of "system-2" fallback for when a "system-1" model is uncertain about its inputs [17].

Generalization to unseen instruction compositions remains a challenging problem. Our hypothesis was that meta-seq2seq is a promising general approach and could be extended to grounded language scenarios by an agent which generates a relevant meta-learning context. Our preliminary results show that such an extension has promise and is an area for future work.

# 6 Acknowledgements

We wish to acknowledge the anonymous reviewers of this work for their helpful feedback. Computational resources were generously provided by the Aalto Science-IT project and CSC – IT Center for Science, Finland. We also acknowledge the the support within the Academy of Finland Flagship programme: Finnish Center for Artificial Intelligence (FCAI).

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

## A  Details on the gSCAN environment

The objects are one of a circle, cylinder or triangle, can be five different sizes and come in the colors red, blue, purple and yellow. The environment is encoded as a grid where each cell is a bag of words, similar to [9].

Available instruction action words are `push`, `pull` and `walk to` and available instruction adverbs are `while spinning`, `while zigzagging`, `hesitantly` and `cautiously`. Actions which the agent must product are in the set of `WALK`, `STAY`, `LTURN`, `RTURN`, `PUSH` and `PULL`.

Split A is an in-distribution validation set split, containing instructions, target objects and target locations which can be found in the training set.

Splits B to F are out-of-distribution sets where the target object has an unseen description made up of combinations of familiar terms (for example red square in split C) or is at a location not seen before during training (for example, southwest of the agent in split D).

Split G is a "meta-learning" split, where an example of "cautiously" is seen only $k$ times during training. We don't consider this split in our work, though initial experiments showed that our approach does not solve this task when the number of related examples is small. There is a good discussion in Ruis and Lake [41] about a data-augmentation method which can help to solve this split.

## B  Conditional Probability Distribution of gSCAN splits

Figure 2: One-step ahead conditional distributions in various gSCAN splits for targets. Shown is $P(a_t|a_{t-1})$ where $a_t$ is down the rows and $a_{t-1}$ is along the columns). Bolded are conditional probabilities notably different from the training split.

(a) Training split

|       | PULL | PUSH | STAY | LTURN | RTURN | WALK |
|-------|------|------|------|-------|-------|------|
| PULL  | 0.52 | 0.00 | 0.33 | 0.00  | 0.00  | 0.15 |
| PUSH  | 0.00 | 0.27 | 0.26 | 0.26  | 0.00  | 0.21 |
| STAY  | 0.19 | 0.05 | 0.00 | 0.00  | 0.00  | 0.75 |
| LTURN | 0.00 | 0.01 | 0.01 | 0.73  | 0.00  | 0.24 |
| RTURN | 0.00 | 0.00 | 0.18 | 0.14  | 0.00  | 0.68 |
| WALK  | 0.00 | 0.00 | 0.20 | 0.35  | 0.17  | 0.28 |

(b) Split A

|       | PULL | PUSH | STAY | LTURN | RTURN | WALK |
|-------|------|------|------|-------|-------|------|
| PULL  | 0.52 | 0.00 | 0.32 | 0.00  | 0.00  | 0.16 |
| PUSH  | 0.00 | 0.28 | 0.25 | 0.27  | 0.00  | 0.21 |
| STAY  | 0.25 | 0.07 | 0.00 | 0.00  | 0.00  | 0.68 |
| LTURN | 0.00 | 0.02 | 0.02 | 0.73  | 0.00  | 0.23 |
| RTURN | 0.00 | 0.00 | 0.20 | 0.13  | 0.00  | 0.67 |
| WALK  | 0.00 | 0.00 | 0.19 | 0.33  | 0.19  | 0.29 |

(c) Split H

|       | PULL | PUSH | STAY | LTURN | RTURN | WALK |
|-------|------|------|------|-------|-------|------|
| PULL  | 0.00 | 0.00 | 0.00 | **1.00** | 0.00 | 0.00 |
| PUSH  | 0.00 | 0.00 | 0.00 | 0.00  | 0.00  | 0.00 |
| STAY  | 0.00 | 0.00 | 0.00 | 0.00  | 0.00  | 0.00 |
| LTURN | **0.08** | 0.00 | 0.00 | 0.78 | 0.00 | 0.14 |
| RTURN | 0.00 | 0.00 | 0.00 | 1.00  | 0.00  | 0.00 |
| WALK  | 0.00 | 0.00 | 0.00 | 0.89  | 0.11  | 0.00 |

(d) Split D

|       | PULL | PUSH | STAY | LTURN | RTURN | WALK |
|-------|------|------|------|-------|-------|------|
| PULL  | 0.52 | 0.00 | 0.33 | 0.00  | 0.00  | 0.15 |
| PUSH  | 0.00 | 0.27 | 0.26 | 0.27  | 0.00  | 0.20 |
| STAY  | 0.16 | 0.05 | 0.00 | 0.00  | 0.00  | 0.79 |
| LTURN | 0.00 | 0.01 | 0.04 | 0.69  | 0.00  | 0.26 |
| RTURN | 0.00 | 0.00 | 0.00 | 0.00  | 0.00  | **1.00** |
| WALK  | 0.00 | 0.00 | 0.18 | 0.53  | 0.07  | 0.22 |

As shown in Figure 2, there are significant conditional distribution shifts between the various gSCAN splits. In the training split, both $P(\text{LTURN}|\text{PULL}) = 0$ and $P(\text{PULL}|\text{LTURN}) = 0$. On the contrary, $P(\text{PULL}|\text{LTURN}) = 1$ in Split H and $P(\text{LTURN}|\text{PULL}) > 0$. In Split D, there is a much higher probability of making a right turn after walking, which corresponds to the agent going southwest (since navigation in gSCAN goes first in the vertical direction, then in the horizontal direction). While these one-step-ahead conditional probabilities don't capture the full picture, since there are other variables such as other previous actions as well as the instruction, they do give a snapshot of the nature of the distribution shift problem in splits D and H.

## C    Training and Model Details

For the both the meta-seq2seq model and Transformer model we use a hidden size of 128 units and fully-connected layer size of 512 units. During training, dropout is not used and weight decay is set to $10^{-3}$ with the AdamW optimizer. Learning rate warmup is used up to step 5000 to a peak learning rate of $10^{-5}$, then decayed on a log-linear schedule from steps 5000 to 50000 to $10^{-6}$. Beta values are left at their defaults, $\beta_1 = 0.9$ and $\beta_2 = 0.999$. Gradients norms are clipped at 0.2 to improve training stability. We use 16-bit precision during training and make use of gradient accumulation in order to simulate large batch sizes where memory is limited.

The Transformer baseline we use is different and more vanilla than the ViLBERT inspired architecture used in [39]. The Transformer follows an encoder-decoder structure, where the state and instruction are jointly encoded and the actions are decoded with causal masking. Each cell in the state is encoded as a bag-of-words and concatenated together, such that different components correspond to different properties of the cell. Learnable 2D positional encodings are appended to each cell. The cells are flattened and concatenated with the position-encoded inputs to form the encoder sequence.

For the meta-seq2seq model, there are Transformer encoders for both instructions and meta-learning action supports. The Transformer encoders for the instructions are tied for both the query instruction and the key support instructions. The Transformer encoder for the instruction encodes the state (as above) and decodes the instruction without any causal masking. An inducing point is appended to the end of the instruction such that the instruction can be represented as a single vector for the purposes of attention computation. Attention is taken between each word in the query instruction and each encoded inducing point for each of the support instructions, then multiplied with the encoded inducing points corresponding to each of the support action sequences. The output sequence of actions is then decoded using a Transformer decoder with causal masking.

## D    Computational Resource Usage and Reproducibility Requirements

Experiments were run on our internal GPU cluster. Running a meta-learning experiment to 30,000 iterations takes about 3 days on a NVIDIA Tesla V100 GPU. For 7 different experiment runs with 10 seeds each, the total compute time is about 210 GPU-days, though in practice the experiments can be run in parallel.

The batch size (4096) we use is quite large and does not fit in GPU memory for a consumer grade GPU. In order to achieve these batch sizes, we use gradient accumulation, so the batch size for each backward step might be 256, but then gradients are averaged over 16 steps to make an effective batch size of 4096. This trades training time for optimization stability.

## E    Code and Datasets

To assist the reader in understanding our work we provide a copy of our PyTorch code for the models, code used to generate figures and tables in this work and experimental code at this URL: `https://neurips-larel-meta-learning-gscan-generalize-submission.s3.eu-north-1.amazonaws.com/submission.zip`

## F    Additional Comparisons

We show additional related work comparisons and hyperparameters in Tables 3 and 4.

The related work of **?** ] reports impressive results on all splits, however we don't compare in this case, since the problem setup the authors used in that work is different from our work and the comparison works. In **?** ] the model predicts *chunks* of actions adverb behaviour as a single vector simultaneously, as opposed to having to autoregressively model the action sequence directly. Therefore, that work does not need to handle the distribution shift problem referred to in Appendix B.

| | seq2seq | GECA | FiLM | RelNet | LCGN | Planning | RD Random/RL | Ours(o) |
|---|---|---|---|---|---|---|---|---|
| | 2020 | 2020 | 2021 | 2021 | 2020 | 2020 | 2022 | Ours |
| A | 97.15 ± 0.46 | 87.6 ± 1.19 | 98.83 ± 0.32 | 97.38 ± 0.33 | 98.6 ± 0.9 | 94.19 ± 0.71 | 98.39 ± 0.17 | 0.95 ± 0.01 |
| B | 30.05 ± 26.76 | 34.92 ± 39.30 | 94.04 ± 7.41 | 49.44 ± 8.19 | 99.08 ± 0.69 | 87.31 ± 4.38 | 62.19 ± 24.08 | 0.96 ± 0.01 |
| C | 29.79 ± 17.70 | 78.77 ± 6.63 | 60.12 ± 8.81 | 19.92 ± 9.84 | 80.31 ± 24.51 | 81.07 ± 10.12 | 56.52 ± 29.70 | 0.97 ± 0.01 |
| D | 0.00 ± 0.00 | 0.00 ± 0.00 | 0.00 ± 0.00 | 0.00 ± 0.00 | 0.16 ± 0.12 | | 43.60 ± 6.05 | 0.35 ± 0.05 |
| E | 37.25 ± 2.85 | 33.19 ± 3.69 | 31.64 ± 1.04 | 42.17 ± 6.22 | 87.32 ± 27.38 | 52.8 ± 9.96 | 53.89 ± 5.39 | 0.85 ± 0.20 |
| F | 94.16 ± 1.25 | 85.99 ± 0.85 | 86.45 ± 6.67 | 96.59 ± 0.94 | 99.33 ± 0.46 | | 95.74 ± 0.75 | 0.97 ± 0.01 |
| H | 19.04 ± 4.08 | 11.83 ± 0.31 | 11.71 ± 2.34 | 18.26 ± 1.24 | 33.6 ± 20.81 | | 21.95 ± 0.03 | 0.86 ± 0.02 |

Table 3: Additional related work comparisons.

| | ViLBERT [39] | Modular [41] | Role-guided [28] | Transformer Ours | Ours(o, A) Ours | Ours(o) Ours |
|---|---|---|---|---|---|---|
| Learning Rate | 0.0015 | 0.001 | 0.001 | 0.0001 | 0.0001 | 0.0001 |
| Batch Size | 128 | 200 | 200 | 4096 | 4096 | 4096 |
| Steps | 114.96K | 73K | 150K | 35K | 35K | 35K |
| #params | 3M | | | 13.2M | 13.2M | 13.2M |

Table 4: Hyperparameters used in the related work comparisons of Table 1

# G Oracle Function

Table 5: Examples of what supports the oracle function generates for a given query instruction and environment. The first two examples are from the training data and the last example is from Split H. Note that we never generate the same instruction as the query instruction in the supports, and we also never generate any Split H instruction in the supports. Also note that in some cases, the environment makes pushing or pulling an object impossible, even though it is in the instruction, see the second row for an example of this.

| Environment | Query Instruction | Target Actions | Supports | |
|---|---|---|---|---|
|  | walk to a small circle hesitantly | LTURN(2) WALK STAY WALK STAY WALK STAY WALK STAY RTURN WALK STAY | push a small circle hesitantly | LTURN LTURN WALK STAY WALK STAY WALK STAY WALK STAY RTURN WALK STAY PUSH STAY PUSH STAY PUSH STAY PUSH STAY |
| | | | pull a small circle hesitantly | LTURN LTURN WALK STAY WALK STAY WALK STAY WALK STAY RTURN WALK STAY pull STAY pull STAY pull STAY pull STAY |
| | | | walk to a small circle while spinning | LTURN(4) LTURN LTURN WALK LTURN(4) WALK LTURN(4) WALK LTURN(4) WALK LTURN(4) RTURN WALK |
| | | | walk to a small circle while zigzagging | LTURN LTURN WALK RTURN WALK LTURN WALK WALK WALK |
| | | | walk to a small circle | LTURN LTURN WALK WALK WALK WALK RTURN WALK |
|  | push a red small cylinder while zigzagging | WALK RTURN WALK LTURN WALK RTURN WALK LTURN WALK PUSH | walk to a red small cylinder while zigzagging | WALK RTURN WALK LTURN WALK RTURN WALK LTURN WALK |
| | | | pull a red small cylinder while zigzagging | WALK RTURN WALK LTURN WALK RTURN WALK LTURN WALK |
| | | | push a red small cylinder while spinning | LTURN(4) WALK LTURN(4) WALK LTURN(4) WALK LTURN(4) RTURN WALK LTURN(4) WALK |
| | | | push a red small cylinder hesitantly | WALK STAY WALK STAY WALK STAY RTURN WALK STAY WALK STAY |
| | | | push a red small cylinder | WALK WALK WALK RTURN WALK WALK |
|  | pull a green big cylinder while spinning | LTURN(4) WALK LTURN(4) LTURN WALK LTURN(4) WALK LTURN(4) WALK LTURN(4) PULL LTURN(4) PULL LTURN(4) PULL LTURN(4) PULL | walk to a green big cylinder while spinning | LTURN(4) WALK LTURN(4) LTURN WALK LTURN(4) WALK LTURN(4) WALK |
| | | | push a green big cylinder while spinning | LTURN(4) WALK LTURN(4) LTURN WALK LTURN(4) WALK LTURN(4) WALK LTURN(4) PUSH LTURN(4) PUSH |
| | | | pull a green big cylinder while zigzagging | WALK LTURN WALK WALK WALK PULL PULL PULL PULL |
| | | | pull a green big cylinder hesitantly | WALK STAY LTURN WALK STAY WALK STAY WALK STAY PULL STAY PULL STAY PULL STAY PULL STAY |
| | | | pull a green big cylinder | WALK LTURN WALK WALK WALK PULL PULL PULL PULL |

The oracle function generates relevant instructions by the use of a templating mechanism, which replaces verbs and adverbs in the sentence with other verbs and adverbs, such that the whole combination is still in distribution, but not the same as the query instruction. The rules of the system are:

- Replace "pull" with "push" and "walk to"

- Replace "walk to" with "push" and "pull" (but not if "while spinning" is the adverb)
- Replace "push" with "walk to" and "pull" (but not if "while spinning" is the adverb)
- Replace "while zigzagging" with "hesitantly", nothing and "while spinning" (but not if "push" is the verb)
- Replace "hesitantly" with "while zigzagging", nothing and "while spinning" (but not if "push" is the verb)
- Replace "while spinning" with "hesitantly", "while zigzagging" and nothing

It is possible that an instruction with the same symbols for `pull ... while spinning` is generated as a query instruction after permutation at training time, however the probability of this happening is low. We measured that for a single pass through the training data, approximately 3% of permuted instructions matched `pull ... while spinning`, 0.3% of the permuted targets matched PULL actions followed by four LTURN instructions, and their intersection was 0.001% of the data.

Three examples of what the oracle function generates for a given query instruction and environment can be found in Table 5.

## H  Dictionaries

| Word | Symbol | Action | Symbol |
|------|--------|--------|--------|
| 'a' | 0 | PULL | 0 |
| 'big' | 1 | PUSH | 1 |
| 'blue' | 2 | STAY | 2 |
| 'cautiously' | 3 | LTURN | 3 |
| 'circle' | 4 | RTURN | 4 |
| 'cylinder' | 5 | WALK | 5 |
| 'green' | 6 | | |
| 'hesitantly' | 7 | | |
| 'pull' | 8 | | |
| 'push | 9 | | |
| 'red' | 10 | | |
| 'small' | 11 | | |
| 'square' | 12 | | |
| 'to' | 13 | | |
| 'walk' | 14 | | |
| 'while spinning' | 15 | | |
| 'while zigzagging' | 16 | | |

Table 6: Default mapping of words and actions to symbols

# I  Permuter Blocks

| Original sentence | Token-symbol permutation | Permuted sentence |
|---|---|---|
| push a red small square hesitantly | 1, 2, 3, 4, 5, 6, 7, 8, 9, 10, 11, 12, 13, 14, 15, 16 | push a red small square hesitantly |
| push a red small square hesitantly | 5, 13, 7, 4, 10, 14, 6, 11, 9, 12, 1, 8, 3, 2, 0, 16, 15 | square cylinder pull big cautiously small |
| walk to a yellow circle while zigzagging | 6, 9, 0, 5, 7, 8, 10, 13, 1, 4, 12, 16, 15, 11, 3, 14, 17, 2 | cautiously small green blue hesitantly yellow |
| push a small circle hesitantly | 10, 3, 7, 5, 4, 17, 8, 6, 13, 12, 1, 15, 11, 0, 2, 16, 9, 14 | square red while spinning circle green |
| walk to a circle while zigzagging | 6, 4, 9, 10, 1, 2, 5, 12, 15, 17, 0, 7, 14, 13, 11, 16, 8, 3 | small to green big pull |
| walk to a green big cylinder | 7, 1, 15, 5, 8, 3, 16, 13, 2, 14, 6, 10, 17, 9, 4, 12, 11, 0 | circle push hesitantly while zigzagging big cautiously |
| pull a big cylinder | 17, 15, 9, 2, 14, 6, 10, 3, 1, 12, 7, 4, 11, 16, 8, 0, 5, 13 | big yellow while spinning green |
| pull a green cylinder while zigzagging | 8, 17, 4, 9, 16, 15, 14, 2, 3, 12, 0, 5, 10, 6, 1, 7, 13, 11 | cautiously pull walk while spinning to |
| pull a big square while zigzagging | 5, 4, 9, 2, 3, 11, 1, 10, 16, 12, 0, 15, 14, 13, 8, 7, 17, 6 | while zigzagging cylinder circle walk yellow |
| pull a small square while zigzagging | 0, 14, 3, 5, 9, 11, 6, 12, 1, 10, 4, 15, 7, 13, 17, 8, 16, 2 | big a while spinning hesitantly while zigzagging |
| pull a small square | 12, 3, 10, 5, 7, 8, 13, 14, 1, 17, 9, 15, 0, 16, 11, 4, 6, 2 | big square while spinning a |
| push a square while spinning | 0, 9, 12, 1, 4, 7, 3, 8, 17, 5, 2, 11, 15, 14, 16, 13, 6, 10 | cylinder a while spinning to |
| walk to a circle hesitantly | 9, 0, 3, 7, 6, 15, 10, 12, 2, 17, 14, 13, 1, 11, 16, 4, 5, 8 | while zigzagging small push green square |
| pull a yellow big circle hesitantly | 11, 3, 6, 14, 17, 2, 7, 15, 9, 0, 1, 12, 13, 5, 10, 8, 16, 4 | push small circle cautiously yellow while spinning |
| walk to a red big circle hesitantly | 11, 8, 2, 9, 12, 0, 13, 17, 6, 14, 1, 10, 16, 15, 3, 4, 5, 7 | cautiously while spinning small big pull square yellow |
| push a yellow cylinder while spinning | 16, 2, 7, 11, 6, 10, 1, 14, 13, 8, 5, 12, 4, 9, 15, 0, 17, 3 | pull while zigzagging cautiously red a |
| walk to a cylinder hesitantly | 11, 17, 8, 3, 12, 2, 16, 5, 9, 10, 7, 13, 0, 4, 1, 14, 6, 15 | big circle small blue cylinder |
| walk to a circle hesitantly | 11, 13, 8, 12, 15, 9, 4, 5, 0, 16, 3, 1, 6, 2, 10, 17, 7, 14 | red blue small while spinning cylinder |
| push a small cylinder hesitantly | 13, 15, 2, 11, 14, 3, 7, 10, 9, 8, 6, 0, 16, 1, 4, 12, 5, 17 | pull to a cautiously red |
| walk to a big circle while spinning | 2, 15, 14, 4, 0, 16, 7, 6, 11, 12, 17, 1, 8, 9, 13, 10, 5, 3 | to push blue while spinning a red |
| push a yellow small cylinder hesitantly | 2, 4, 3, 15, 12, 17, 13, 14, 16, 6, 11, 9, 10, 5, 7, 0, 8, 1 | green blue big push yellow walk |
| push a red big cylinder | 17, 16, 10, 8, 4, 13, 0, 2, 9, 7, 1, 12, 11, 15, 3, 6, 5, 14 | hesitantly yellow big while zigzagging to |
| walk to a red circle hesitantly | 12, 17, 16, 15, 0, 3, 7, 9, 11, 14, 1, 4, 13, 6, 5, 10, 8, 2 | cylinder green square big a push |
| push a yellow cylinder while zigzagging | 0, 17, 11, 14, 10, 5, 13, 7, 6, 15, 16, 12, 3, 2, 8, 1, 4, 9 | while spinning a push cylinder circle |
| walk to a yellow circle hesitantly | 10, 8, 12, 13, 0, 3, 6, 1, 15, 2, 11, 14, 7, 16, 4, 9, 5, 17 | circle while zigzagging red yellow a big |
| pull a red small cylinder hesitantly | 10, 4, 16, 17, 5, 14, 12, 2, 11, 7, 0, 3, 15, 8, 9, 6, 13, 1 | small red a cautiously walk blue |
| pull a small cylinder while zigzagging | 0, 16, 2, 9, 11, 8, 1, 5, 10, 12, 17, 13, 6, 7, 4, 3, 14, 15 | red a to pull walk |

Table 7:  Instructions and possible mapping permutations generated by the permuter block.

| Original actions | Permutation | Permuted actions |
|---|---|---|
| WALK LTURN WALK(2) | 1, 2, 3, 4, 5, 6 | WALK PUSH WALK(2) |
| WALK LTURN WALK(2) | 3, 2, 4, 1, 0, 5 | WALK PUSH WALK(2) |
| LTURN(6) WALK LTURN(4) WALK LTURN(4) WALK LTURN(4) RTURN WALK LTURN(4) WALK LTURN(4) WALK | 2, 0, 1, 5, 3, 4 | WALK(6) RTURN WALK(4) RTURN WALK(4) RTURN WALK(4) LTURN RTURN WALK(4) RTURN WALK(4) RTURN |
| WALK(5) RTURN WALK(2) PULL(3) | 3, 2, 0, 1, 4, 5 | WALK(5) RTURN WALK(2) LTURN(3) |
| WALK STAY LTURN WALK STAY WALK STAY PULL STAY PULL STAY PULL STAY PULL STAY | 5, 3, 4, 2, 0, 1 | PUSH RTURN STAY PUSH RTURN PUSH RTURN WALK RTURN WALK RTURN WALK RTURN WALK RTURN |
| WALK STAY RTURN WALK STAY | 5, 0, 3, 2, 1, 4 | RTURN LTURN PUSH RTURN LTURN |
| WALK(4) LTURN WALK(5) | 0, 5, 1, 2, 3, 4 | RTURN(4) STAY RTURN(5) |
| LTURN(4) WALK LTURN(4) WALK LTURN(4) RTURN WALK | 4, 5, 3, 0, 1, 2 | PULL(4) STAY PULL(4) STAY PULL(4) PUSH STAY |
| LTURN(2) WALK STAY WALK STAY RTURN WALK STAY WALK STAY WALK STAY WALK STAY | 0, 4, 5, 2, 3, 1 | STAY(2) PUSH WALK PUSH WALK LTURN PUSH WALK PUSH WALK PUSH WALK PUSH WALK |
| LTURN(2) WALK RTURN WALK | 3, 0, 1, 2, 5, 4 | STAY(2) RTURN WALK RTURN |
| LTURN(4) RTURN WALK LTURN(4) WALK LTURN(4) WALK LTURN(4) WALK LTURN(4) PUSH | 0, 3, 2, 4, 1, 5 | RTURN(4) PUSH WALK RTURN(4) WALK RTURN(4) WALK RTURN(4) WALK RTURN(4) LTURN |
| LTURN(2) WALK RTURN WALK LTURN WALK RTURN WALK LTURN WALK RTURN WALK LTURN WALK RTURN WALK LTURN WALK PULL(2) | 0, 4, 5, 2, 3, 1 | STAY(2) PUSH LTURN PUSH STAY PUSH LTURN PUSH STAY PUSH LTURN PUSH STAY PUSH LTURN PUSH STAY PUSH PULL(2) |
| WALK STAY WALK STAY WALK STAY WALK STAY LTURN WALK STAY | 1, 0, 4, 2, 5, 3 | LTURN RTURN LTURN RTURN LTURN RTURN LTURN RTURN STAY LTURN RTURN |

Table 8:  Actions and possible mapping permutations generated by the permuter block.

The permuter block shuffles the indices mapping words to symbols in the dictionary given in Table 6. Tables 7 and 8 give an example of how the permuted sequences might look to the encoders. Essentially the individual symbols no longer hold any special meaning without reference to the demonstrations, only conditional autoregressive probabilities up to a permutation hold meaning.

# J  Failure case analysis

We also studied the remaining failure cases on Split H for the best-case version of our model, to see where the remaining challenges are. To do this, we ran the model autoregressively on its instructions and supports without any causally-masked teacher forcing. We observed four types of failure:

Figure 3: Failure case analysis. In (a) we classify failure cases for the meta-seq2seq model. Note that multiple failures can happen in a single example, so percentages do not add to 100. In (b) and (c), we show the edit distance frequency distribution, as well as edit distance as a function of the number of `PULL` instructions in the target sequence. Models that generalize poorly will have a larger edit distance for more complex target instructions

| Failure reason | % of all failure cases |
|---|---|
| Did not turn | 78.82% |
| Spurious Pull | 33.1% |
| Missed Pull | 8.09% |
| Other reason | 0.058% |

(a) Failure case classification

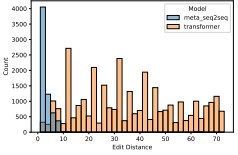

(b) Edit-distance freq. dist.

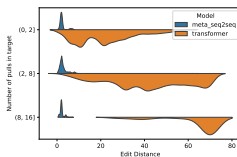

(c) Pulls vs Edit Distance dist.

- **Did not turn** The agent "missed" a turn instruction when generating an instruction path that requires one, eg, because the target object is not in the same row as the agent. In this case, `WALK WALK` is generated as opposed to `LTURN WALK` or `RTURN WALK`.

- **Spurious Pull** The agent generates a `PULL` instruction where it should not generate one.

- **Missed Pull** The agent does not generate a `PULL` instruction where it should generate one.

- **Other reason** The failure is more complex or multi-faceted than can be attributed to the above reasons.

The majority of failures can be attributed to "**Did not turn**". To compute whether the agent is erroneously picking `WALK` but is uncertain, we compute the mean entropy of the prediction logits in all cases where there is a failure to turn. A high amount of uncertainty should correspond to 0.5. The mean value of the entropy plus standard deviation in this case is $0.22 \pm 0.24$. This indicates that the agent is somewhat certain about its decision, but there are cases where it is completely certain (in error) or quite uncertain, where further training may improve the situation.

There are also a few cases where the agent generates a `PULL` instruction or does not generate one when it is expected to ("**Spurious Pull**" and "**Missed Pull**"). We hypothesized that this may be because of an asymmetry between the actions seen for "push while spinning" in the context and "pull while spinning" in the target (the number of `PUSH` or `PULL` actions can differ depending the location of the target object and its surroundings), but we didn't see any concrete relationship here.

We also analyzed the edit distances between the target sequence and the predicted sequences from the model. Because of a lack of teacher-forcing, we expect that the number of exact matches will be about the same, but it is possible that one error could cause compounding errors if the transformer decoder inputs effectively become out of distribution. In practice we found that the edit distances follow a power law. In the majority of cases, only a small number (1 to 3) errors are made throughout the whole sequence, and only in a small number of cases do we make a large number of errors. Since the length of the novel sequences the model must generate can be quite large, this indicates that the errors made by the model don't come down to a lack of generalization due to poor fitting. We would expect the edit distances to be larger if generalization was the issue. The following example may be illustrative. Say for example the target is "pull while spinning" on an object that needs to be pulled three times. The target target sequence would need to contain `LTURN(4) PULL LTURN(4) PULL LTURN(4) PULL`. A model that failed to generalize would generate in the best case "push while spinning" with targets ending in `LTURN(4) PUSH LTURN(4) PUSH LTURN(4) PUSH` (edit distance of 3), or "pull", with target ending in `PULL PULL PULL` (edit distance of 12). But we see that an order of magnitude more cases have a shorter edit distance in Figure 3, so the issue is not fundamentally down to a lack of generalization.

