# OpenReview forum: "Meta-learning from demonstrations improves compositional generalization"
_NeurIPS.cc/2022/Workshop/LaReL — LaReL 2022_

### Official Review · Reviewer_QZsS · 2022-10-16

**Rating:** 7
**Confidence:** 4

**Review:**

This paper applies the meta-seq-to-seq approach of Lake 2019 (which in the era of big LMs we might call "in-context learning") to the gSCAN task of Ruis et al. (I'm guessing that the model is trained via behavior cloning rather than some other IL / RL technique, though it's not clear from the paper.) Unlike in past work, this paper studies a "few-shot" rather than "zero-shot" generalization setting, because the input to the meta-learner contains examples of target concepts being used; nevertheless, past work has found these few-shot problems quite challenging too. This approach achieves very strong results on the gSCAN suite, performing particularly well on splits D & H.

I think this is a nice, self-contained, and convincingly evaluated result, and would be a great fit for the workshop. If the authors wish to continue in this direction towards a full conference paper, there are a few things that could be done to strengthen the presentation:

- Direct comparison to other methods with equivalent data conditions. If the oracle is able to generate e.g. 2 examples of a new concept / concept combination being used, then it seems fair to make that same set of examples available to the baseline (e.g. just adding them into the training set). I suspect this will not change the top-level results of this paper at all, but will make the claim that the meta-learning component is important more convincing.

- Less hand-coding in the oracle. There's been some past work on slightly more automated ways of calculating the "support set" for meta-learning models like this one (e.g https://arxiv.org/abs/2010.03706, https://arxiv.org/abs/2112.08696) and it would be interesting to see if those work here. In the few-shot setting, is it possible to automatically identify the informative examples?

- Better use of interactivity. The main difference between this paper and the earlier paper by Lake is the interactive environment, and it would be great to see that used more explicitly, e.g. by doing inverse RL or some mix of supervised learning and RL. These kinds of example-based meta-learning algorithms are much less studied in MDPs than they are in ordinary supervised learning problems, and this paper seems like a great jumping-off point for investigating them in more detail.

---

### Official Review · Reviewer_QD76 · 2022-10-18

**Rating:** 7
**Confidence:** 4

**Review:**

This work is about using a meta sequence-to-sequence learning approach to improve performance on unseen compositions of instructions in a grounded environment. The authors propose to extend meta-seq2seq to work on the gSCAN embodied environment and conduct several experiments to compared their approach to other recent works on gSCAN.

I found the paper well-written and clear. Compositional generalization of language instructions is a very important research topic. gSCAN remains a challenging benchmark for NLP and RL, especially some of the splits as mentioned in the paper. The authors did a great job to adapting the meta-seq2seq to gSCAN and show improvements. They also mention the current limitation of the extension, i.e. the use of an oracle function to generate the instructions and actions. I'm interested in seeing future work on replacing the oracle with a generative model. I recommend this paper.

---

### Decision · Program_Chairs · 2022-10-20

Accept